# HyPHEN: A Hybrid Packing Method and Optimizations for Homomorphic Encryption Based Neural Network

## Abstract

Private Inference (PI) enables users to enjoy secure AI inference services while companies comply with regulations. Fully Homomorphic Encryption (FHE) based Convolutional Neural Network (CNN) inference is promising as users can offload the whole computation process to the server while protecting the privacy of sensitive data. Recent advances in AI research have enabled HE-friendly deep CNN like ResNet. However, FHE-based CNN (HCNN) suffers from high computational overhead. Prior HCNN approaches rely on dense packing techniques that aggregate as many channels into the ciphertext to reduce element-wise operations like multiplication and bootstrapping. However, these approaches require performing an excessive amount of homomorphic rotations to accumulate channels and maintain dense data organization, which takes up most of the runtime. To overcome this limitation, we present HyPHEN, a deep HCNN implementation that drastically reduces the number of homomorphic rotations. HyPHEN leverages two convolution algorithms, CAConv and RAConv. Alternating between two convolution algorithms leads to a significant reduction in rotation count. Furthermore, we propose hybrid gap packing method for HyPHEN, which gathers sparse convolution results into a dense data organization with a marginal increase in the number of rotations. HyPHEN explores the trade-off between the computational costs of rotations and other operations, and finds the optimal point minimizing the execution time. With these optimizations, HyPHEN takes 3.4-4.4× less execution time than the state-of-the-art HCNN implementation and brings the runtimes of ResNet on CIFAR10 inference down to 1.44-13.37s using a GPU-accelerated HEAAN library.

## 1 Introduction

Private inference (PI) has recently gained the spotlight in the MLaaS domain as cloud companies should comply with privacy regulations such as GDPR Regulation (2016) and HIPAA Act (1996). PI enables inference services at the cloud server while protecting the privacy of the client and the intellectual properties of the service provider. For instance, hospitals can provide a private medical diagnosis of diseases, and security companies can provide private surveillance systems without accessing client's sensitive data (Kumar et al., 2020; Bowditch et al., 2020).

PI can be achieved using various cryptographic primitives (Gentry, 2009; Yao, 1982; Costan & Devadas, 2016). Fully Homomorphic Encryption (FHE), which is a set of cryptographic schemes that can directly evaluate a rich set of functions on encrypted data, is especially suited for PI. FHE-based PI solution uniquely features 1) full offloading of the computation process to the server, 2) succinct data communication requirement, and 3) non-disclosure of any information about the model except the inference result. Such benefits have driven researchers to investigate convolutional neural network (CNN) PI implementations using FHE (Gilad-Bachrach et al., 2016; Brutzkus et al., 2019; Dathathri et al., 2020; Lee et al., 2022a; Aharoni et al., 2020).

To implement CNN using FHE, activation functions should be replaced with polynomials as FHE only supports arithmetic operations of addition and multiplication. Given the constraint, two classes of polynomial activation functions have been proposed: (i) low-degree polynomials (Gilad-Bachrach

et al., 2016; Chabanne et al., 2017) replacing the activation functions in training neural networks, and (ii) more precise high-degree approximation of ReLU (Lee et al., 2021) that replaces RELU during PI without additional training. However, both approaches lack practicality; low-degree polynomials are not applicable to deep neural networks and high-degree approximation significantly degrades the runtime of PI. Recently, Park et al. (2022) showed that deep homomorphic CNNs (HCNNs) can be trained with low-degree polynomials even for complex image datasets with their proposal, AESPA, which utilizes orthogonal polynomial bases and fuses activation functions with batch normalization (BN) to turn them into second-degree polynomials. AESPA does not sacrifice runtime nor accuracy unlike prior approaches, thus we employ AESPA in our work.

Another line of research lies in implementing an efficient convolution algorithm in FHE. Gazelle (Juvekar et al., 2018) proposed a convolution algorithm that can compute a single Conv layer on FHE. However, Gazelle's method cannot be directly applied to continuous convolutions as it requires adjusting arrangement of data by re-encrypting ciphertexts after every Conv layer. Lee et al. (2022a) modified Gazelle's convolution by densely mapping data into a ciphertext before entering the next Conv layer. However, the current state of HCNN is far from being practical. Using the convolution algorithm of Lee et al. (2022a) and approximated ReLU, inference times of ResNet20 CIFAR-10 are 1662/174s using a single/64 threads in our CPU environment. Despite the unique advantages of FHE-based PI, the huge runtime overhead prevents FHE from being the go-to solution for PI.

We propose **Hy**brid **P**acking method and optimizations for **H**omomophic **E**ncryption-based neural **N**etwork (HyPHEN), which mitigates the huge overhead of HCNN with an optimized convolution algorithm and packing method. We observe that after AEPSA is applied, rotation operations in HCNN take up the majority of the runtime (See Appendix A) and most of the rotations (92-99%) are spent to implement the sum of channels within the same ciphertext and maintain data organization. We design a novel convolution algorithm named RAConv that does not require rotations to accumulate channels. In addition, based on the observation that maintaining a single data organization necessitates massive unnecessary rotations, we design RAConv to take the new data organization based on the replication of the images. By alternating between two data organizations, we remove rotations priorly required to adjust the data organization. HyPHEN also includes a novel *Hybrid Packing* (HP) method that effectively handles the gap arising from strided convolution (Section 3.2). HyPHEN achieves 39.6s and 1.44s of runtime in ResNet20 for the CIFAR-10 dataset on CPU and GPU, respectively. The key contributions of the paper are as follows:

- We propose a replication-based convolution method, RAConv, that can effectively reduce two types of unnecessary rotations which are the major bottleneck in HCNN.
- We propose a novel hybrid packing (HP) method that can utilize the entire slots of a ciphertext with a marginal increase in the number of rotations.
- Our experiments show that our HCNN implementation with HyPHEN improves the inference latency by 3.4-4.4$\times$ over prior state-of-the-art HCNNs for ResNet on CIFAR-10.

## 2 BACKGROUND

### 2.1 FULLY HOMOMORPHIC ENCRYPTION

FHE is a set of public key encryption schemes that can perform computation on encrypted data. Among several popular FHE schemes, RNS-CKKS (Cheon et al., 2018) has been broadly adopted in the PI domain as it supports fixed-point numbers and *slot batching*. A *plaintext* in RNS-CKKS is an unencrypted degree-$N$ polynomial in a cyclotomic polynomial ring, $R_Q = \mathbb{Z}_Q[X]/(X^N + 1)$. A plaintext maps to a message which is a vector of $N/2$ real (or complex) numbers. Thus a single plaintext batches $N/2$ *slots*, which can store complex or real numbers. CKKS encrypts a plaintext into a *ciphertext* in $R_Q^2$. $Q$ is a ring modulus which is represented by a set of prime modulus obtained from the Chinese Remainder Theorem (CRT) as $\prod_{i=0}^{l} q_i$ ($1 \leq l \leq L$). $L$ and $l$ denote the initial and current *level* of a ciphertext. The level is an HE-specific resource that determines the number of multiplications applicable to a given ciphertext. We also denote the associated level of ring modulus using subscript as $Q_L$ or $Q_l$. We denote the plaintext and ciphertext of a message $\mathbf{a}$ as $\langle\mathbf{a}\rangle$ and $[\mathbf{a}]$. HE operations of addition, multiplication, and rotation can be described as follows:

- HE.Eval([$\mathbf{a}$],[$\mathbf{b}$],$f_l$) = HE.Eval([$\mathbf{a}$],$\langle\mathbf{b}\rangle$,$f_l$) = [$f_l(\mathbf{a},\mathbf{b})$]
- HE.Rotate([$\mathbf{a}$],r) = [rot($\mathbf{a}$,r)]

$f_l$ denotes linear operations, either Hadamard addition or multiplication. rot($\mathbf{a}$,r) represents cyclically shifting vector $\mathbf{a}$ by r to the left. Unlike addition and rotation, multiplication in RNS-CKKS requires additional rescale operation, which consumes a level by dividing $ct \in R_{Q_l}$ into $ct' \in R_{Q_{l-1}}$. If a ciphertext has no level left after a series of multiplications, *bootstrapping* (Bossuat et al., 2022) is needed to reconcile the levels and allow further operation. Bootstrapping, the most costly operation in HE, consists of multiple HE operations including rescale operations. After bootstrapping, the level of the resulting ciphertext becomes $L' = (L - L_b)$ where $L_b$ is the depth of rescale operations in the bootstrapping circuit. As it is beneficial to perform many operations before bootstrapping, $L$ should be sufficiently larger than $L_b$. However, large $L$ decreases the *security level*, which should be high enough to tolerate cryptographic attacks. The security level is roughly proportional to $N/L$. Considering the security requirement of HE, large $L$ requires large $N$ ($\geq 2^{15}$). Thus prior works on FHE (Bossuat et al., 2021; Jung et al., 2021; Lee et al., 2022b) target $N = 2^{15}$ to $2^{17}$.

Table 1 shows the execution time of HE operations on a system specified in Section 4.1. We measured the execution time of each operation at the initial level (max level) of a ciphertext and thus the execution time may decrease for ciphertexts with lower levels. Bootstrapping consumes over two orders of magnitude longer runtime than other operations, but boostrapping does not occur as frequently as others. Except for bootstrapping, Rotate and MulCt are the most time-consuming operations in HE, which is due to the expensive key-switching procedure.

| Benchmark | AddPt | AddCt | MulPt | MulCt | Rescale | Rotate | Boot |
|-----------|-------|-------|-------|-------|---------|--------|------|
| Time (ms) | 0.572 | 0.202 | 0.506 | 17.301 | 3.904 | 15.492 | 2156.605 |

Table 1: The benchmark of HE operations averaged over 100 iterations on CPU (64 threads). Pt and Ct postfixes represent ciphertext-plaintext and ciphertext-ciphertext operation, respectively.

## 2.2 CONVOLUTION ON HOMOMORPHIC ENCRYPTION

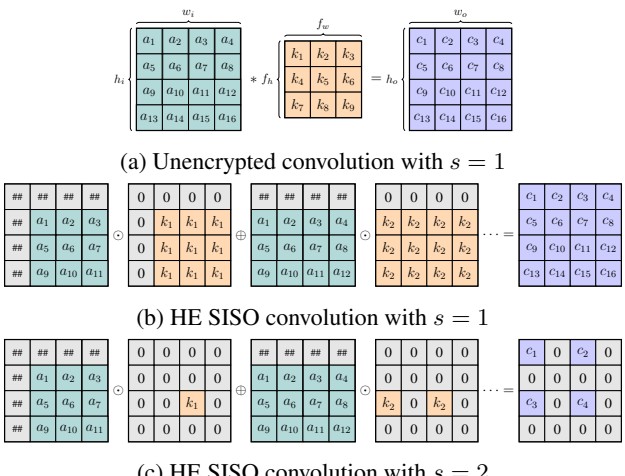

(a) Unencrypted convolution with $s = 1$

(b) HE SISO convolution with $s = 1$

(c) HE SISO convolution with $s = 2$

Figure 1: SISO convolution. ciphertexts and plaintexts are illustrated as a 2D matrix but are actually stored in 1D manner with each matrix row concatenated.

In this subsection, we describe previous convolution algorithms for FHE. We represent input and output tensors with tuples $\{w_i, h_i, c_i\}$ and $\{w_o, h_o, c_o\}$, and Conv layers with $\{f_w, f_h, c_i, c_o\}$. We denote the stride of the convolution as $s$ and assume $padding = 1$ for simplicity. Gazelle (Juvekar et al., 2018) proposed efficient *SISO (single-input and single-output channel)* convolution algorithms on HE. Figures 1b, 1c show SISO convolutions for $s = 1, 2$. Filter elements are separated into $f_w f_h$ plaintexts. Each slot in $i$-th plaintext stores $k_i$ or 0 (punctured) depending on whether $k_i$ is involved in the computation of the output pixel at the same slot. SISO operates as follows: 1) rotate an encrypted input image with different indexes according to plaintext filter, 2) perform Hadamard multiplication, and 3) accumulate the multiplication results to obtain the output. Alternatively, if we

prepare the filter plaintexts inversely rotated and directly multiply them with the input ciphertext, the rotations can be performed after MulPt operations (Zhang et al., 2021), which we dub *lazy-SISO*. We selectively use SISO and lazy-SISO to minimize rotations for convolution. Detailed explanation of lazy-SISO is provided in Appendix E

In more general cases of multiple channels, convolution on HE is performed in a SIMD manner. If the size of a channel image is smaller than the number of slots in a ciphertext, multiple channels can be placed in a ciphertext. For example, if $slot = 2^{15}$ and an input channel is sized $w_i h_i = 32 \times 32$ as in the input image of the CIFAR-10 dataset, $\frac{slot}{w_i h_i} = 32$ channels can be placed in a single ciphertext in an aligned manner (i.e. *channel-aligned*). Then, the process of convolution for a channel-aligned input ciphertext storing multiple channels can be described as follows. Suppose $c_i = \frac{slot}{w_i h_i}$. First, SISO is performed on $c_i$ input channels in a SIMD manner (see Figure 2c), which produces $c_i c_o$ convolution outputs $MK^{(i,j)}$ ($1 \leq i \leq c_i$, $1 \leq j \leq c_o$). To compute the result for the $k$-th output channel, $\sum_{i=1}^{c_i} MK^{(i,k)}$ is accumulated by *RaS (Rotate and Sum)*, which is repeated until all the output channels are acquired. Finally, the ciphertexts packed with output channels are realigned to match the next layer's input alignment by *IR (Image Realign)*. Throughout this paper, we refer to this convolution that takes a channel-aligned ciphertext as the input as *channel-aligned convolution (CAConv)*. CAConv can be further optimized for the case where the input tensor is not large enough to fill all the slots in a ciphertext; $\frac{slot}{w_i h_i c_i}$ repeated copies of the input tensor are placed in a ciphertext, then $\frac{slot}{w_i h_i c_i}$ output channels can be computed together in a single ciphertext (Lee et al., 2022a).

Strided convolution ($s > 1$) using SISO algorithm creates a gap between valid values (see Figure 1c). A ciphertext with a gap underutilizes its slots, leading to throughput degradation. While Juvekar et al. (2018) can remove the gap by a client-aided re-encryption process, non-interactive PI shall remove the gap through masking and rotation, which incur additional rotation overhead and also consume more levels. Lee et al. (2022a) proposed a multiplexed packing method that can be combined with CAConv (*MP-CAConv*) to mitigate the overheads. In the IR stage of MP-CAConv, multiplexed packing fills the gap with other channels (see Figure 3b), which we refer to as the *repacking process*. Other than the repacking process, MP-CAConv is very similar to CAConv. In MP-CAConv, IR collectively refers to realigning and repacking process.

Tile tensoring based convolution proposed in (Aharoni et al., 2020) is a yet another convolution algorithm for FHE. While tile tensoring based convolution can be efficient alternative to SISO-based convolution when the image size is sufficiently high, our paper mainly focuses on SISO-based convolution, which can be applied more broadly.

In RNS-CKKS, the *hoisting* optimization allows multiple rotations to share common sub-operations composing a rotation operation. First-level hoisting shares the front decomposition sub-operations when rotating a single ciphertext multiple times with different indexes. Second-level hoisting shares the rear ModDown sub-operations when rotating and summing up multiple ciphertexts. We refer the reader to Bossuat et al. (2021) for further details of the hoisting optimization. First-level hoisting can be applied to SISO, whereas second-level hoisting can be applied to lazy-SISO.

## 2.3 POLYNOMIAL ACTIVATION FUNCTION ON HOMOMORPHIC ENCRYPTION

Non-linear activation functions, such as ReLU, cannot be used in HCNN. They must be replaced with polynomial functions as HE only supports addition and multiplication operations. To directly replace ReLU with approximate polynomials, approximation error should be negligible over a wide range to retain the original accuracy of a CNN model. Lee et al. (2021) designed a precise approximation of ReLU having an $l_1$ norm approximation error less than $2^{-13}$ in the range of [-50, 50], obtained by a composition of $\{15, 15, 27\}$ degree polynomials. This approximation-based approach has a benefit that it can be applied to pretrained neural networks. However, evaluation of high-degree polynomials imposes a significant runtime overhead in HCNN inference. Another approach is to train neural networks with low-degree polynomial activations as in Ishiyama et al. (2020); Chabanne et al. (2017); Obla et al. (2020); Hesamifard et al. (2019); Thaine et al. (2019). While this approach requires retraining, operational cost is much cheaper compared to high-degree polynomials. Recently AESPA (Park et al., 2022) demonstrated that CNN trained with low-degree polynomials can achieve equivalent accuracy to ReLU-based networks across various CNN networks and image datasets. AESPA replaces batch normalization (BN) and ReLU with the composition of orthogonal basis polynomials and basis-wise BN as follows:

$$f(x) = \gamma \sum_{i=0}^{d} \hat{f}_i \frac{h_i(x) - \mu}{\sqrt{\sigma^2 + \epsilon}} + \beta \qquad (1)$$

Here, $h_i$ are the orthogonal bases, $\sigma$ and $\mu$ are the mean and variance computed from BN, and $\gamma$ and $\beta$ are trainable parameters. For $d = 2$, AESPA turns into a second-degree polynomial with different coefficients for each channel on inference. We adopt AESPA in this paper, which leads to better runtime performance.

## 2.4 THREAT MODEL

We adopt the same threat model as prior PI works. A client sends encrypted data to an untrusted server. The server performs CNN inference using HE operations and returns inference results to the client. The client decrypts the resulting ciphertext to obtain the private result. The server only holds the client's public keys and cannot decrypt any intermediate ciphertexts in the inference process. The client does not know any information about the processing at the server other than the result.

## 3 METHOD

### 3.1 REPLICATION-ALIGNED CONVOLUTION

The main performance bottleneck of CAConv is the massive number of rotations. CAConv requires an enormous number of rotations to implement the sum of the channels within the same ciphertext (RaS) and the relocation of the channels (IR) to match the next layer's input representation. These rotations take up most of the time in CAConv. For example, when $N = 2^{16}$, rotations for RaS and IR account for 49% and 43% of the total number of rotations in ResNet20, respectively. Furthermore, IR consumes an additional level for masking to extract the values.

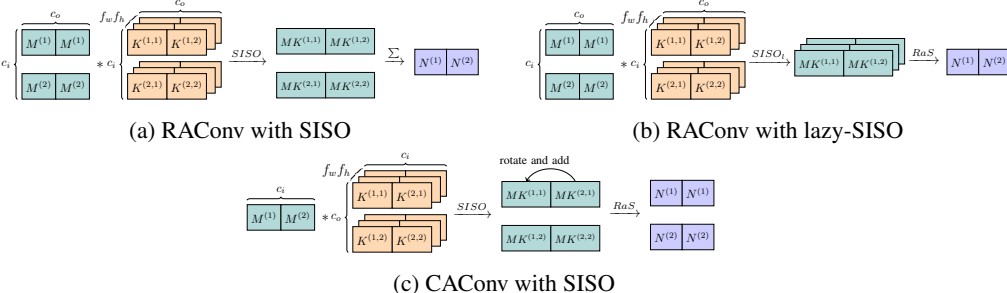

Figure 2: CAConv and RAConv. The single superscript denotes channel and the superscript pair denotes (input channel, output channel). We simplify the notation of $M^{(a)}K^{(a,b)}$ as $MK^{(a,b)}$.

To mitigate the performance bottleneck caused by rotations in CAConv, we design *Replication-Aligned Convolution (RAConv)* to receive the alternative data representation. In CAConv, the output ciphertext of RaS contains replications of the channel sum (i.e. *replication-aligned*) as shown in Figure 2c. RAConv receives replication-aligned ciphertexts as input and skips IR and RaS stages. Figure 2a shows an example of performing RAConv. RAConv takes $c_i$ input ciphertexts each filled with replications of a single input channel and weight plaintexts aligned in output channel order. RAConv operates as follows: 1) perform parallel SISO, which outputs $c_i$ ciphertexts where the $i$-th ciphertext contains $MK^{(i,j)}$ for all $j$ values, and 2) accumulate the ciphertexts by simple HE additions. SISO in RAConv actually increases the number of rotations because $c_i$ input ciphertexts require $c_i(f_w f_h - 1)$ rotations during parallel SISO. Instead, we utilize lazy-SISO with RAConv, which requires much fewer $(f_w f_h - 1)$ rotations for SISO. RAConv produces a densely-packed channel-aligned ciphertext that complies with the CAConv input data organization, so we alternate between RAConv and CAConv. RAConv-CAConv chain halves the RaS rotations previously required in two CAConvs to $c_i log(c_o)$.

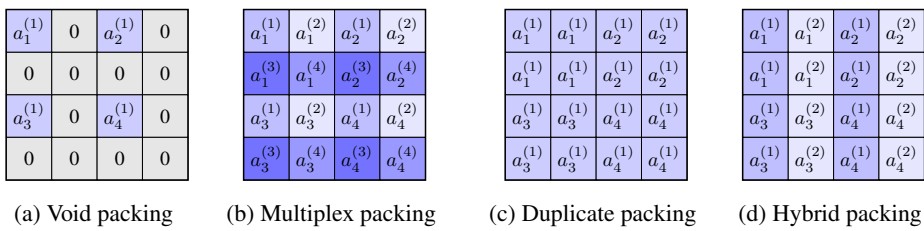

Figure 3: Comparison of gap packing methods to fill gap induced by downsampling layers.

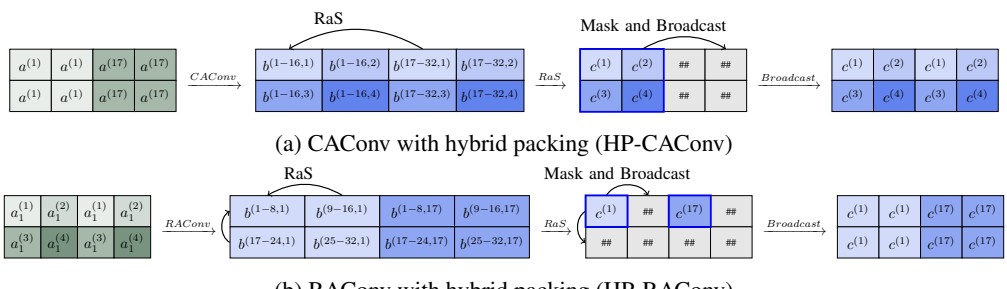

Figure 4: The procedure of CAConv and RAConv with hybrid packing

## 3.2 HYBRID PACKING

SISO convolution suffers from low slot utilization in ciphertexts due to two reasons. First, strided convolution creates a gap between valid values (see Figure 3a). Second, the small size of input tensors creates empty slots in ciphertexts. Due to the security requirement of FHE, the number of slots in a ciphertext is often larger than the size of an input tensor. Underutilization of slots in SISO leads to severe throughput degradation in HCNN.

Prior state-of-the-art HCNN implementation (MP-CAConv) mitigates underutilization of slots using multiplexed packing and input repetition (see Section 2.2). However, they cause a lot of additional rotation overhead to adjust the data organization. In MP-CAConv, RaS operation is used to accumulate SISO results in a ciphertext. Under input repetition, RaS operation returns an output channel at the slots where an input ciphertext stores the first channel of an input tensor. Multiplexed channels are also accumulated through RaS operation, thus values constituting an output channel exist only in non-gap slots, as shown in Figure 3a. To restore the data organization of MP-CAConv, invalid values are masked off and empty slots are filled with other channels through IR process. In the presence of input repetition, IR spends $O(c_o)$ rotations to relocate output channels and $\frac{slot}{w_o h_o c_o}$ additional rotations for generating input repetition.

To reduce the relocating overhead between convolutional layers, we propose a novel image packing method, *hybrid packing (HP)*. HP fills the gap with channel duplicates of multiple channels (See Figure 3d). We design HP based on two key observations. First, applying convolution over a duplicate-packed ciphertext (Figure 3c) produces a multiplex-packed output ciphertext as in Figure 3b. Second, converting a void-packed ciphertext into a duplicate-packed ciphertext requires fewer rotations than converting it into a multiplex-packed ciphertext. Duplicate packing only needs $O(\log(gap_{size}))$ rotations while multiplexed packing requires $O(gap_{size})$ rotations.

HP is a hybrid of duplicate packing and multiplexed packing. We represent a hybrid-packed ciphertext by a pair of numbers, the number of multiplexed channels $m$, and the number of duplicates $d$. For example, Figure 3d shows $(m, d) = (2, 2)$ HP. Packing of the ciphertext switches between two HP settings while processing CAConv and RAConv as shown in Figure 4. We denote HP parameter $(m, d)$ of the input and output ciphertext as $(m_{in}, d_{in})$ and $(m_{out}, d_{out})$. Input repetition is no longer required as HP with larger $d_{in}$ can be used instead. Duplicates of HP produce different output channels within the gap (See $c^{(1)}$ and $c^{(17)}$ in Figure 4b) Then, the IR process adjusts the output ciphertext's organization from $(m_{out}, d_{out}) = (d_{in}, 1)$ to $(d_{in}, m_{in})$, which only requires $\mathcal{O}(\log m_{in})$

| Method | $ct_{in}$ | $ct_{out}$ | SISO | RaS | IR |
|---|---|---|---|---|---|
| **(Lee et al., 2022a)** | $\lceil \frac{w_i h_i c_i}{n} \rceil$ | $\lceil \frac{w_o h_o c_o}{n} \rceil$ | $ct_{in}(f_w f_h - 1)$ | $\frac{w_i h_i c_i c_o}{n} \log c_i$ | $c_o + \log \frac{n}{w_o h_o c_o}$ |
| **HP-CAConv** | $\frac{w_i h_i c_i d_{in}}{n}$ | $\frac{c_o}{d_{in}}$ | $ct_{min}(f_w f_h - 1)$ | $ct_{out} \log \frac{c_i}{ct_{in}}$ | $ct_{out} \log m_{in}$ |
| **HP-RAConv** | $\frac{c_i}{m_{in}}$ | $\frac{w_i h_i c_o m_{in}}{n}$ | $ct_{min}(f_w f_h - 1)$ | $ct_{out} \log m_{in}$ | $ct_{out} \log m_{in}$ |

Table 2: The rotation complexity of the convolutions. We denote the numbers of input and output ciphertexts as $ct_{in}, ct_{out}$. Then, $ct_{min} = \min(ct_{in}, ct_{out})$ considering SISO and lazy-SISO.

rotations, which fill the gaps. After performing a series of CAConv and RAConv, the HP organization of the output ciphertext returns to the initial $(m_i n, d_i n)$. The complete procedures of RAConv and CAConv with HP are described more in detail in Appendix H.

Compared to MP-CAConv, HP significantly reduces the rotations in RaS and IR. The rotation complexities of MP-CAConv (Lee et al., 2022a) and our hybrid-packed convolutions are shown in Table 2. For both HP convolution methods, the product of the numbers of input and output ciphertexts remains constant ($ct_{in} \cdot ct_{out} = \frac{w_i h_i c_i c_o}{n}$). Compared to MP-CAConv, the number of rotations for RaS is reduced by about $ct_{in}$ times for both HP-CAConv and HP-RAConv. IR stage of HP repacks the gap with duplicates. Rotation decreases from $c_o$ of MP-CAConv to mere $\log m_{in}$ rotations per output ciphertext.

HP convolutions require more rotations than MP-CAConv for SISO; $ct_{in}$ of MP-CAConv is always smaller than or equal to $ct_{min}$ of HP-CAConv or HP-RAConv. Nevertheless, hoisting can be applied to SISO which reduces the significance of the rotation cost of SISO, and the reduction of rotations in RaS and IR overwhelms the increase in SISO.

All things considered, HP reduces the overall number of rotations required for convolutions. We can also explore various combinations for the $(m, d)$ pair to minimize the total number of rotations. The choice of $(m, d)$ decides $ct_{in}$ and $ct_{out}$ values and creates a trade-off between SISO, RaS, and IR costs, and also affects the number of ciphertexts we have to perform bootstrapping with. We provide an in-depth performance analysis with regard to the choice of $(m, d)$ in Section 4.2.

### 3.3 The ResNet Architecture on HyPHEN

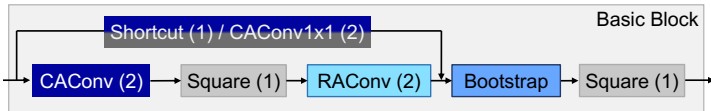

Figure 5: The structure of ResNet basic block built on HyPHEN. The level consumption per block is written in parentheses. In the downsampling block, pointwise convolution is added to the critical path. Otherwise, a simple shortcut is added.

HyPHEN combines RAConv and HP to build the entire CNN model. Figure 5 shows the basic block of ResNet implemented on HyPHEN. There are three more considerations when deciding the order of operations. First, bootstrapping is cheaper when placed after RAConv, and not CAConv, because the number of ciphertexts is smaller at the moment. Second, to match the level between the shortcut path and the main CAConv-RAConv path, bootstrapping should be placed either before residual connections diverge or after they converge. Last, it is beneficial to perform convolutional layers at the lowest level possible. The complexity of FHE operations such as rotation is proportional to the level $l$ of ciphertext. Therefore, the lower the calculated level, the smaller the computational cost.

All things put together, our ResNet basic block implementation consumes a total of 6 levels. The level consumption of each layer is represented in the parenthesis of each block. CAConv and RAConv use HP and consume one level for each SISO and IR. Activation uses AESPA and consumes one level. AESPA is a quadratic polynomial having different coefficients for each channel while training. During inference, we fuse the coefficients into nearby layers, then the activation becomes a simple square function $x^2$. We set the initial ciphertext level $L$ to six and perform bootstrapping when the level becomes zero.

# 4 EVALUATION

## 4.1 EXPERIMENTAL SETUP

We ran HCNN inference on CPU and GPU environments using the RNS-CKKS library, HEAAN. CPU instance is equipped with AMD EPYC 7452 running at 2.35GHz (64 cores) and 480GB DRAM. GPU experiments are conducted at the same system with an additional NVIDIA A100 GPU with 80GB HBM. Our HCNN inference experiments use the CIFAR-10 (Krizhevsky et al., 2009). We evaluate ResNet20/32/44/18 trained with AESPA on PyTorch and applied the fusion technique to all the networks. Our RNS-CKKS parameters satisfy 128-bit security level (Cheon et al., 2019) with polynomial degree $N = 2^{16}$ and hamming weight 192. Multiplication and bootstrapping primes each occupy 48 bits and over 56 bits, respectively. The bootstrapping implementation consumes 17 levels in our setup.

## 4.2 OPTIMAL POINT

We explore the optimal $(m, d)$ pair for HP to minimize the latency of ResNet 20/18. Table 3 shows the operation count of rotation and bootstrapping, which are major contributors to the runtime. Rotation is categorized into SISO and non-SISO. Only SISO rotations can be optimized with hoisting. Table 3 presents an instance of MP-CAConv and three (or two) instances of our HP-based convolution. Instances of HP-based convolution consist of settings with minimal bootstrapping, minimal rotations, and optimal latency. In ResNet20, The optimal point corresponds to the minimal bootstrapping. In ResNet18, the amount of rotations hikes following the channel increment. The optimal point sacrifices bootstrapping counts to play fewer rotations. We opt for these two settings to evaluate in the following section. A more thorough parameter study on GPU is presented in Appendix B.

| Model | Packing | (m,d) | | | | Rotations | | | | Boot | CPU Runtime(s) |
| | | L1 | L2 | L3 | L4 | SISO | RaS | IR | total | | |
|---|---|---|---|---|---|---|---|---|---|---|---|
| ResNet20 | MP-CAConv | | | | | 152 | 924 | 800 | 1876 | 11 | 74.3 ± 0.8 |
| | Optimal | (1,2) | (2,4) | (4,8) | - | 152 | 580 | 187 | 919 | 11 | **39.6 ± 0.3** |
| | Min Rot | (1,2) | (1,8) | (2,16) | - | 240 | 407 | 142 | 789 | 16 | 46.3 ± 0.3 |
| ResNet18 | MP-CAConv | | | | | 176 | 19472 | 4787 | 24435 | 12 | 558.6 ± 5.6 |
| | Min Boot | (1,1) | (4,1) | (8,2) | (16,4) | 184 | 12678 | 3380 | 16242 | 12 | 366.5 ± 7.2 |
| | Optimal | (1,1) | (1,4) | (2,8) | (4,16) | 448 | 3976 | 1131 | 5555 | 27 | **185.5 ± 2.0** |
| | Min Rot | (1,1) | (1,4) | (1,16) | (2,32) | 672 | 3376 | 1007 | 5055 | 39 | 195.0 ± 1.7 |

Table 3: Runtime for the ResNet instances with different $(m, d)$ parameters and packing strategies.

## 4.3 SENSITIVITY STUDY

In Figure 6, three activation function & convolution algorithm pairs are evaluated. Set1 follows the implementation of Lee et al. (2022a), using AppReLU with MP-CAConv. Set1 spends most of the time for bootstrapping in ResNet20 inference. Set2 employs the square activation of AESPA along with MP-CAConv. AESPA reduces the number of levels consumed by activation functions, thus requiring much less time for bootstrapping. Set2 experiences $5.1\times$ and $2.9\times$ reduction in bootstrapping time in ResNet20 and ResNet18, respectively. For both networks, most of the runtime is spent on CAConv in Set2.

By introducing HyPHEN in Set3, the execution times spent for convolutional layers are reduced by $3.1\times$ and $5.0\times$ in ResNet20 and ResNet18, respectively. The impact of Hy-PHEN is amplified for more complex networks like ResNet18; HyPHEN resolves the problem

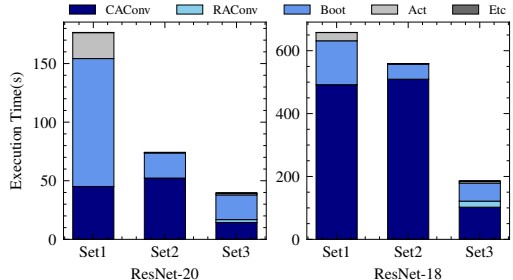

Figure 6: The execution time of ResNet20/18 with three settings on our CPU environment. Set1 uses appReLU with MP-CAConv (Lee et al., 2022a). Set2 uses AESPA with MP-CAConv and Set3 represents HyPHEN, using AESPA with our convolution method. FHE parameters used in the experiments are explained in Table 11

of MP-CAConv showing a superlinear increase in the number of rotations as the number of channels increases. Overall, HyPHEN achieves 3.4-4.4× speedup compared to the prior state-of-the-art implementation of Lee et al. (2022a) (Set1).

## 4.4 EXECUTION TIME

Table 4 shows the runtimes of various ResNet instances on CIFAR-10. We measured the execution time of running inference with a single CIFAR-10 image in our CPU/GPU environments. Our ResNet20/32/44 implementations on GPU take merely a few seconds to complete. Furthermore, We demonstrate running ResNet18 for the first time. As ResNet18 has 4× more channels than ResNet20/32/44, execution time largely depends on the convolutional layer. Table 4 again demonstrates that RAConv effectively reduces the overall runtime of the conv layer, as our operation count analysis in Table 3. Detailed comparison with Lee et al. (2022a) is provided in Appendix A

| Execution time (s) | CPU (64 cores) | | | | GPU | | | |
|---|---|---|---|---|---|---|---|---|
| | ResNet20 | ResNet32 | ResNet44 | ResNet18 | ResNet20 | ResNet32 | ResNet44 | ResNet18 |
| **HP-CAConv** | 14.43 | 20.96 | 26.68 | 102.34 | 0.49 | 0.70 | 0.90 | 7.06 |
| **HP-RAConv** | 2.41 | 4.12 | 5.83 | 19.18 | 0.07 | 0.11 | 0.15 | 1.03 |
| **Bootstrap** | 21.08 | 33.93 | 46.07 | 57.53 | 0.82 | 1.31 | 1.80 | 3.49 |
| **Activation** | 1.44 | 2.50 | 3.40 | 6.16 | 0.05 | 0.09 | 0.12 | 0.58 |
| **Etc** | 0.22 | 0.25 | 0.40 | 0.31 | <0.01 | <0.01 | 0.01 | 1.21 |
| **total** | $39.58 \pm 0.3$ | $61.76 \pm 0.7$ | $82.38 \pm 0.9$ | $185.52 \pm 2.0$ | $1.44 \pm 0.02$ | $2.21 \pm 0.02$ | $2.98 \pm 0.02$ | $13.37 \pm 0.09$ |

Table 4: HyPHEN Inference time of a single CIFAR-10 image using ResNet models on CPU and GPU. As FC layer and Pooling have a tiny execution time, we gather them at Etc.

## 4.5 ACCURACY

In Table 5, we measured inference accuracies for CIFAR-10 images running ResNet model on our RNS-CKKS-based implementation. Though we found an error below the second decimal place at the classifier result, we did not observe any deterioration in the accuracy of ResNet20/32/44. ResNet18 shows -0.08% degradation in accuracy, which is smaller compared to the accuracy drop in Lee et al. (2022a).

| Top-1 Acc (%) | ResNet20 | ResNet32 | ResNet44 | ResNet18 |
|---|---|---|---|---|
| **Backbone** | 92.18 | 93.36 | 94.04 | 95.1 |
| **Measured** | 92.17 | 93.35 | 94.08 | 95.02 |

Table 5: Comparison of the inference accuracies for the CIFAR-10 dataset running ResNet models on our RNS-CKKS-based implementation.

The difference in accuracy drop can be explained by whether the original network is executed as is or an approximation has been made.

## 5 LIMITATIONS

ResNet networks are the only models addressed in the paper. Broader experiments toward various models such as Liu et al. (2022) would show the practicality of FHE-based PI more clearly.

## 6 CONCLUSION

In this paper, we proposed an efficient convolution algorithm RAConv, and a novel packing method Hybrid Packing. We showed HyPHEN, FHE-based ResNet architecture implementation applying proposed optimizations. Our experiments on real machine show 3.4-4.4× lower latency for ResNet20/32/44/18 compared to Lee et al. (2022a). Using GPU acceleration, HyPHEN demonstrates 1.44/2.21s/2.98s/13.37s execution time for running ResNet20/32/44/18 on CIFAR-10 dataset.

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

## A    BOTTLENECK ANALYSIS

Figure 7 shows the runtime analysis of the convolutional layer, which is conducted on actual layer instances of ResNet20. In Lee et al. (2022a), rotation accounts for 83-94% of the total convolution time, which is reduced to 46-77% with our method. The optimized convolutional layer in HyPHEN leads to a smaller ratio of rotation in the total execution time. Table 6 shows runtime breakdown of each operation and detailed comparison with (Lee et al., 2022a) in our CPU environment. As we set both implementations to use AESPA with the same HE parameter set, the speedup is solely due to the different packing schemes. In ResNet20, our implementation shows slight increase in activation function runtime. However, latency improvements in convolution layer leads to the $1.87\times$ lower total execution time. Similarly in ResNet18, our implementation reports increased runtime in activation function and bootstrapping time, but $5.77\times$ lower convolution time again leads to $3.33\times$ lower total execution execution time.

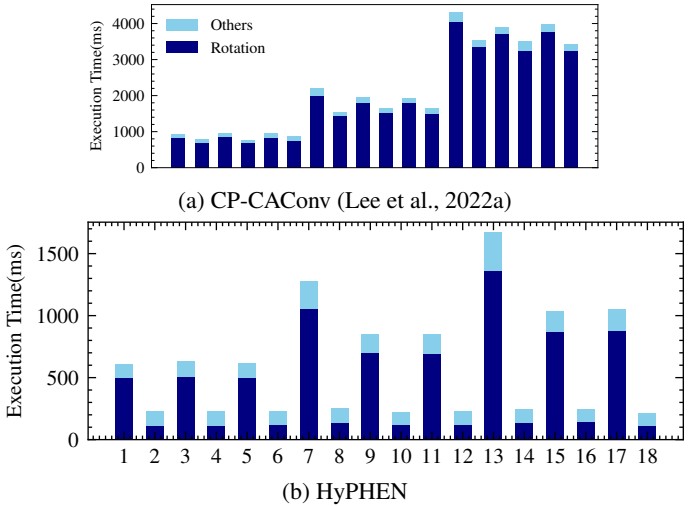

(a) CP-CAConv (Lee et al., 2022a)

(b) HyPHEN

Figure 7: Comparison of rotation time in ResNet20 convolutional layers

| Execution time (s) | ResNet 20 | | | | ResNet18 | | | |
| | Ours | | Lee et al. (2022a) | | Ours | | Lee et al. (2022a) | |
| | runtime | percent | runtime | percent | runtime | percent | runtime | percent |
|---|---|---|---|---|---|---|---|---|
| **CAConv** | 14.43 | 36.4% | 52.32 | 70.4% | 102.33 | 55.2% | 590.53 | 95.9% |
| **RAConv** | 2.41 | 6% | - | - | 19.18 | 10.3% | - | - |
| **Bootstrap** | 21.08 | 53.2% | 21.41 | 28.8 % | 57.53 | 31.0% | 25.91 | 4.2% |
| **Activation** | 1.44 | 3.6% | 0.34 | <0.01% | 6.16 | 3.3% | 0.43 | <0.01 % |
| **Etc** | 0.22 | 0.01% | 0.22 | <0.01% | 0.31 | <0.01% | 0.24 | <0.01 % |
| **Total** | 39.58 ± 0.3 | 100% | 74.29 ± 0.8 | 100% | 185.52 ± 2.04 | 100% | 617.11± 2.04 | 100% |

Table 6: HyPHEN inference time of a single CIFAR-10 image using ResNet models on CPU. As FC layer and Pooling have a tiny execution time, we gather them at Etc.

## B    PARAMETER STUDY

We present a parameter study to explore the optimal HP setting which minimizes latency. Table 7 and 8 show the rotation and bootstrapping counts with varying (m, d) available in ResNet20 and 18 and the execution time running the network on GPU. We only represent the $(m, d)$ pair of CA-Conv, as $m$ and $d$ are exchanged at RAConv. In ResNet20, we start with $(m, d) = (1, 2)$ to remove input repetition as the size of the input tensor in the first layer ($32 \times 32 \times 16$) is smaller than the ciphertext slots ($2^{15}$). Nevertheless, Larger $d$ has not been considered as it leads to more bootstrapping, as shown in our proposed architecture (See Figure 5). As the input ciphertexts go through the downsampling layer, $m \cdot d$ gets quadrupled and the size of the intermediate tensor gets halved. In ResNet20, HP that doubles d every downsampling layer yields optimal performance, which reduces rotation without increasing bootstrap. In ResNet18, the impact of bootstrapping increment

is often smaller than the impact of rotation decrement. The optimal HP setting requires 15 more bootstrappings and 10687 fewer rotations than the minimum bootstrapping HP setting.

| (m,d) | | | Rotations | | | | Boot | GPU Runtime(s) |
|---|---|---|---|---|---|---|---|---|
| L1 | L2 | L3 | SISO | RaS | IR | total | | |
| **(1,2)** | **(2,4)** | **(4,8)** | **152** | **580** | **187** | **919** | **11** | **1.44 ± 0.02** |
| (1,2) | (2,4) | (2,16) | 192 | 539 | 162 | 893 | 13 | 1.56± 0.02 |
| (1,2) | (1,8) | (4,8) | 200 | 448 | 165 | 813 | 14 | 1.59± 0.02 |
| (1,2) | (1,8) | (2,16) | 240 | 407 | 142 | 789 | 16 | 1.71± 0.02 |
| (1,2) | (1,8) | (1,32) | 240 | 419 | 154 | 831 | 20 | 2.06 ± 0.02 |

Table 7: Comparison of the instances of CAConv (m,d) parameters in ResNet20.

| (m,d) | | | | Rotations | | | | Boot | GPU Runtime(s) |
|---|---|---|---|---|---|---|---|---|---|
| L1 | L2 | L3 | L4 | SISO | RaS | IR | total | | |
| (1,1) | (4,1) | (8,2) | (16,4) | 184 | 12678 | 3380 | 16242 | 12 | 18.5 ± 0.09 |
| (1,1) | (4,1) | (8,2) | (8,8) | 208 | 12046 | 2793 | 15047 | 13 | 17.7 ± 0.12 |
| (1,1) | (4,1) | (4,4) | (16,4) | 216 | 9804 | 2786 | 12806 | 14 | 16.9 ± 0.15 |
| (1,1) | (4,1) | (4,4) | (8,8) | 240 | 9172 | 2207 | 11619 | 15 | 15.8 ± 0.12 |
| (1,1) | (4,1) | (4,4) | (4,16) | 288 | 8904 | 1979 | 11171 | 17 | 15.7 ± 0.09 |
| (1,1) | (2,2) | (8,2) | (16,4) | 216 | 10122 | 2920 | 13258 | 14 | 17.4 ± 0.11 |
| (1,1) | (2,2) | (8,2) | (8,8) | 240 | 9490 | 2333 | 12063 | 15 | 17.0 ± 0.20 |
| (1,1) | (2,2) | (4,4) | (16,4) | 248 | 7248 | 2342 | 9838 | 16 | 15.3 ± 0.12 |
| (1,1) | (2,2) | (4,4) | (8,8) | 272 | 6616 | 1763 | 8651 | 17 | 14.7 ± 0.11 |
| (1,1) | (2,2) | (4,4) | (4,16) | 320 | 6348 | 1535 | 8203 | 19 | 14.9 ± 0.14 |
| (1,1) | (2,2) | (2,8) | (8,8) | 336 | 5352 | 1531 | 7219 | 21 | 14.6 ± 0.14 |
| (1,1) | (2,2) | (2,8) | (4,16) | 384 | 5084 | 1311 | 6779 | 23 | 14.1 ± 0.11 |
| (1,1) | (2,2) | (2,8) | (2,32) | 480 | 5004 | 1243 | 6727 | 27 | 14.3 ± 0.13 |
| (1,1) | (1,4) | (4,4) | (16,4) | 312 | 6140 | 2146 | 8598 | 20 | 15.1 ± 0.14 |
| (1,1) | (1,4) | (4,4) | (8,8) | 336 | 5508 | 1567 | 7411 | 21 | 14.0 ± 0.14 |
| (1,1) | (1,4) | (4,4) | (4,16) | 384 | 5240 | 1339 | 6963 | 23 | 13.9 ± 0.10 |
| (1,1) | (1,4) | (2,8) | (8,8) | 400 | 4244 | 1351 | 5995 | 25 | 13.6 ± 0.15 |
| **(1,1)** | **(1,4)** | **(2,8)** | **(4,16)** | **448** | **3976** | **1131** | **5555** | **27** | **13.4 ± 0.09** |
| (1,1) | (1,4) | (2,8) | (2,32) | 544 | 3896 | 1063 | 5503 | 31 | 13.7 ± 0.10 |
| (1,1) | (1,4) | (1,16) | (4,16) | 576 | 3456 | 1067 | 5099 | 35 | 13.4 ± 0.10 |
| (1,1) | (1,4) | (1,16) | (2,32) | 672 | 3376 | 1007 | 5055 | 39 | 13.8 ± 0.11 |
| (1,1) | (1,4) | (1,16) | (1,64) | 672 | 3432 | 1031 | 5135 | 47 | 14.4 ± 0.10 |

Table 8: Comparison of the instances of CAConv (m,d) parameters in ResNet18.

## C  RESNET ARCHITECTURE AND PARAMETERS

Figure 8 presents model architecture of modifed ResNet20 used in HCNN evaluation. Table 9 and 10 shows parameters used in convolution layer of ResNet20/32/44/18. All the parameters $(c_i, c_o, w_i, h_i, w_o, h_o, f_w, f_h, s)$ are determined following the origianl ResNet paper (He et al., 2016).

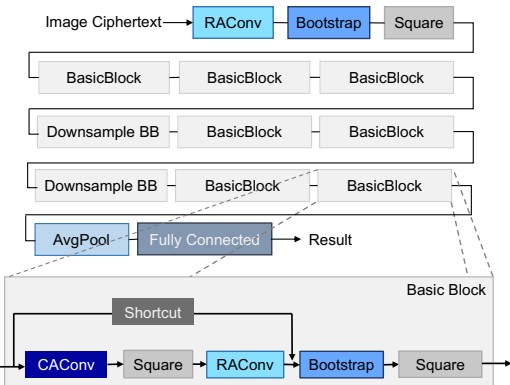

Figure 8: The ResNet20 structure of HyPHEN.

| | Layer1 | Layer2 | | | Layer3 | | |
| | conv | dsconv | pconv | conv | dsconv | pconv | conv |
|---|---|---|---|---|---|---|---|
| $c_i$ | 16 | 16 | 16 | 32 | 32 | 32 | 64 |
| $c_o$ | 16 | 32 | 32 | 32 | 64 | 64 | 64 |
| $w_i(=h_i)$ | 32 | 32 | 32 | 16 | 16 | 16 | 8 |
| $w_o(=h_o)$ | 32 | 16 | 16 | 16 | 8 | 8 | 8 |
| $f_w(=f_h)$ | 3 | 3 | 1 | 3 | 3 | 1 | 3 |
| $s$ | 1 | 2 | 2 | 1 | 2 | 2 | 1 |

Table 9: Parameters used in the convolution layers of ResNet20/32/44

| | Layer1 | Layer2 | | | Layer3 | | | Layer4 | | |
| | conv | dsconv | pconv | conv | dsconv | pconv | conv | dsconv | pconv | conv |
|---|---|---|---|---|---|---|---|---|---|---|
| $c_i$ | 64 | 64 | 64 | 128 | 128 | 128 | 256 | 256 | 256 | 512 |
| $c_o$ | 64 | 128 | 128 | 128 | 256 | 256 | 256 | 512 | 512 | 512 |
| $w_i(=h_i)$ | 32 | 32 | 32 | 16 | 16 | 8 | 8 | 8 | 4 | 4 |
| $w_o(=h_o)$ | 32 | 16 | 16 | 16 | 8 | 8 | 8 | 4 | 4 | 4 |
| $f_w(=f_h)$ | 3 | 3 | 1 | 3 | 3 | 1 | 3 | 3 | 1 | 3 |
| $s$ | 1 | 2 | 2 | 1 | 2 | 2 | 1 | 2 | 2 | 1 |

Table 10: Parameters used in the convolution layers of ResNet18

## D    TRAINING DETAILS

Models used in this paper is all trained using PyTorch (Paszke et al., 2019). For ResNet18 and 20, our training settings are mostly identical to the AESPA; To be specific, networks are trained for 200 epochs using SGD optimizer, 0.1 initial learning rate, 100 batch size, 0.0005 weight decay and 0.9 momentum, and cosine annealing scheduler. We also use soft labels as in (Park et al., 2022) to get higher accuracy. For ResNet32 and 44, we use knowledge distillation (Hinton et al., 2015) to enhance the accuracy, using pretrained ResNet32/44 with 93.4% and 94.1% accuracies as teacher models. As the FC layer of the student and teacher network is identical, teacher's FC layer is directly reused in student network. We trained the student networks by minimizing $l_2$ loss ($L_{kd} = \|f_t - f_s\|_2^2$). ResNet32 and 44 are trained for 200 epochs using SGD optimizer, 0.0005 weight decay and 0.9 momentum. We use 0.05 initial learning rate and learning rate scheduler decays learning rate on epochs 150, 180, 200 by 0.1.

## E    LAZY-SISO

Unlike the original SISO convolution which rotates input ciphertexts before multiplying with filter plaintexts, lazy-SISO proposed in (Zhang et al., 2021) uses inversely rotated filter plaintexts to multiply with input ciphertexts. The actual process of lazy-SISO when $c_i, c_o = 1$ is depicted in Figure 9. After multiplying with filter plaintext, postponed rotation is performed to accumulate intermediate ciphertexts. During this process, multiple ciphertexts sharing the same rotation index are grouped to be accumulated first and then rotated, reducing the amount of rotations. (e.g. $c_i$ ciphertexts are grouped in Figure 2b ). Lazy-SISO is beneficial when input channels, which are to be accumulated, are distributed in different ciphertexts as in RAConv.

## F    MEMORY REQUIREMENT

Memory requirement for HCNN depends on FHE parameters and data representations (packing schemes). In FHE, expansion of data size occurs while encryption and encoding procedure. Resulting ciphertexts and plaintexts are typically orders of magnitude larger than messages. The table 11 shows the actual size of the ciphertext, plaintext and evaluation key on three FHE parameter settings. dnum denotes RNS-decomposition number introduced in (Han & Ki, 2019). Given N, the degree of a cyclotomic polynomial ring, large dnum increases L, max level. `ParamSet1` is used in `Set1` in Figure 6 to reproduce Lee et al. (2022a). As `Set1` uses approximated ReLU for activation, ParamSet1 adopts maximum dnum to have $L = 16$. `ParamSet2` is set with the smallest $L$ among three and is only used in `Set2` on ResNet18. `ParamSet3` is the parameter set used to evaluate

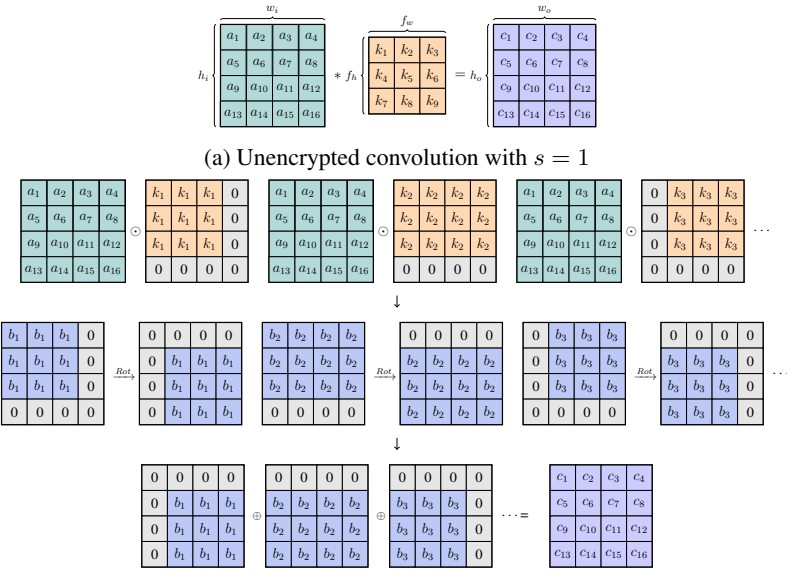

(a) Unencrypted convolution with $s = 1$

(b) HE lazy-SISO convolution with $s = 1$

Figure 9: Lazy-SISO convolution.

HyPHEN and `Set2` on ResNet20 which has $L = 6$. Some FHE operations such as MulCt, Rotate and Conjugate requires specific key switching procedure. Eval key denotes public key used at this process. The size of single Eval key is 2147, 206, 176 MB in ParamSet1, 2 and 3, respectively. To support bootstrapping operation, one relinearization key for MulCt, one conjugate key and 48 rotation keys per rotation index are required. We additionally load frequently used rotation keys to perform convolution. For instance, We load 68 unique Eval keys in ResNet20 which take up 146, 14, 12 GB in ParamSet1, 2, and 3.

Once FHE parameter is determined, the packing scheme determines the number of ciphertext, plaintext to run each ResNet block. Table 12 and 13 shows the required number and total memory size of ciphertexts and plaintexts. We further explain the actual computation procedure to explain how the results are obtained in Appendix H. In SISO-based HCNN kernel, the size of filter plaintexts increases by factor of $w_i h_i$ as each filter element is duplicated to the size of input image, which requires total $f_w f_h w_i h_i c_i c_o$ slots for weight plaintexts. In consequence, weight plaintexts take up the majority of memory regardless of the ciphertext packing method. In table 12, our implementation shows up to 14.75% memory overhead compared to Lee et al. (2022a), which is primarily due to the increase in the number of ciphertexts and bias plaintext. In Table 13, our implementation shows up to 36.8% memory overhead compared to Lee et al. (2022a). The larger memory overhead is caused by using different FHE parameters; In ResNet18, we use ParamSet3 for our implementation and ParamSet2 for Lee et al. (2022a). When using the same ParamSet3, overhead is reduced by up to 7%.

Unlike CPU, GPU memory capacity is more constrained by the current HBM technology to support high bandwidth. As the GPU memory is not capable of loading the weights of the entire model considering ResNet18, weight plaintexts should be loaded separately; while one stream computes the current ResNet block, another stream is overlapped to load next block's weight plaintext. Through profiling the workloads of CPU and GPU activities through NVIDIA Nsight Systems (nvi, 2021), we find that all the copy stream is completed before the end of the compute stream, meaning that loading weight plaintexts does not affect overall execution time. As such, for running large neural networks, fine-grained multi-streaming can be applied to relieve memory capacity constraint.

## G IMAGENET EXPERIMENT

We conducted additional experiments to evaluate the ResNet18 model on the ImageNet dataset. The runtime is 81.85 seconds in our GPU environment. We slightly modified the first pooling layer to

|  | L | dnum | Ciphertext (MB) | Plaintext (MB) | Eval Key (MB) | Total Keys(GB) |
|---|---|---|---|---|---|---|
| **ParamSet1** | 16 | 32 | 17.82 | 8.91 | 2147.48 | 146.03 |
| **ParamSet2** | 3 | 7 | 7.34 | 3.67 | 205.52 | 13.98 |
| **ParamSet3** | 6 | 6 | 10.48 | 5.24 | 176.16 | 11.98 |

Table 11: FHE parameter settings. dnum is tuned to support 16, 3, 6 levels required in `Set1`, `Set2`, `Set3`. Each Ciphertext and Plaintext memory size is represented when current level $l = L$.

| ResNet20 | Layer1 BB | | Layer2 DSB | | BB | | Layer3 DSB | | BB | |
|---|---|---|---|---|---|---|---|---|---|---|
|  | Ours | Lee | Ours | Lee | Ours | Lee | Ours | Lee | Ours | Lee |
| **filter ptxts** | 144 | 144 | 232 | 232 | 144 | 144 | 232 | 232 | 144 | 144 |
| **input ctxts** | 1 | 1 | 1 | 1 | 1 | 1 | 1 | 1 | 1 | 1 |
| **peak ctxts** | 19 | 10 | 19 | 10 | 19 | 10 | 19 | 10 | 19 | 10 |
| **total size (GB)** | 0.70 | 0.68 | 1.18 | 1.07 | 0.79 | 0.69 | 1.19 | 1.08 | 0.79 | 0.71 |
| **memory overhead** | 0.3% | | 10.5% | | 14.7% | | 10.1% | | 12.0% | |

Table 12: Total memory size and the number of each object in ResNet20. We abbreviated Down-sampling Block and Basic Block to DSB and BB.

average pool with kernel size 2×2 and stride 2. As shown in Figure 10, the network receives input ciphertexts processed with modified im2col. Original im2col would transform 224×224×3 images into 147×12544 matrix. As HCNN prefers images size to be exponential of 2, 12544 columns turns to 16384 columns. We further split 16384 columns with stride 2 ($s_{avg}$ in Figure 10) to perform average pooling without rotation, resulting 4096 columns. We pack $slot/4096 = 8$ rows into a ciphertext. As the total number of row is $f_h f_w c_i = 147$, the number of input ciphertext is $\lceil 147/8 \rceil \times 4$. After the initial convolution layer, the number of intermediate ciphertexts becomes 64 and each ciphertext stores 8 channels with (m,d) = 1. Table 14 shows the runtime of ResNet18 on ImageNet. During the experiment, swap memory is used to make up the lack of CPU memory. Loading weight plaintexts from swap memory, which is in SSD, incurs runtime hikes as Table 14 shows large runtimes for Etc.

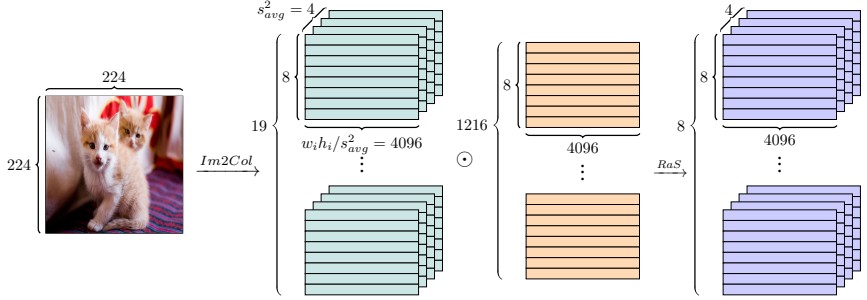

Figure 10: ImageNet Im2Col

## H  Complete convolution procedure of hybrid packed tensor

Figure 11 and 12 show the complete procedure of CAConv and RAConv with HP shown briefly in Figure 4. Both CAConv and RAConv perform the sequence of $\{SISO, RaS, IR\}$ to the input ciphertexts. As in Figure 2, a single superscript represents the channel of input images, and a superscript pair represents (input channel, output channel) of filters. If multiple channels are stored in a box, we represent the list of channels using $\&$ or the range of channels using $-$. We set $c_i, c_o = 32$ and $w_i, h_i = 16$ as the layer 2 of ResNet20. In HP-CAConv (Figure 11), the HP setting of input ciphertext is $(m, d) = (2, 4)$. In HP-RAConv (Figure 12), the HP setting of input ciphertext is $(m, d) = (4, 2)$.

We use different brightness of color to fill the ciphertexts and plaintexts to reflect the actual computation process. In Figure 11 and 12, the intermediate ciphertexts of the CAConv and input ciphertexts of the RAConv are $\frac{c_o}{4}$ and $\frac{c_i}{4}$ times larger than input ciphertext of the CAConv, respec-

| ResNet18 | Layer1 | | Layer2 | | | | Layer3 | | | | Layer4 | | | |
| | BB | | DSB | | BB | | DSB | | BB | | DSB | | BB | |
| | Ours | Lee | Ours | Lee | Ours | Lee | Ours | Lee | Ours | Lee | Ours | Lee | Ours | Lee |
|---|---|---|---|---|---|---|---|---|---|---|---|---|---|---|
| filter ptxts | 2304 | 2304 | 3712 | 3712 | 2304 | 2304 | 3712 | 3712 | 2304 | 2304 | 3712 | 3712 | 2304 | 2304 |
| input ctxts | 2 | 2 | 2 | 2 | 4 | 1 | 4 | 1 | 4 | 1 | 4 | 1 | 4 | 1 |
| max ctxts | 38 | 19 | 38 | 19 | 76 | 10 | 76 | 10 | 76 | 10 | 76 | 10 | 76 | 10 |
| total size (GB) | 7.27 | 7.39 | 14.14 | 11.81 | 8.67 | 7.33 | 16.09 | 11.76 | 9.84 | 7.33 | 16.10 | 11.76 | 9.84 | 7.34 |
| memory overhead | -0.2% | | 19.7% | | 18.3% | | 36.8% | | 34.3% | | 36.8% | | 34.2% | |

Table 13: Total memory size and the number of each object in ResNet18. DSB and BB refers to downsampling Block and basic block.

| | Im2Col | CAConv | RAConv | Bootstrap | Activation | Etc | total |
|---|---|---|---|---|---|---|---|
| **runtime** | 3.99 | 5.98 | 2.68 | 28.97 | 0.45 | 39.78 | 81.85 ±1.99 |
| **percent** | 4.9 % | 7.3 % | 3.3 % | 35.4% | 0.5% | 48.60% | 100% |

Table 14: HyPHEN Inference time of a single ImageNet image using ResNet18 models on GPU.

tively. The CAConv, activation and RAConv in a ResNet block is processed at once to mitigate the huge number of intermediate ciphertexts. To avoid an increase of memory footprint, the operation on the input ciphertext continues until the ciphertext shrinks again. Thus, the tuple of operations $\{SISO, RaS, IR, Square, SISO_l\}$ are processed to an input ciphertext and then accumulated. Blocks colored with high brightness show actual working set, which means $\frac{c_o}{4}$ times larger intermediate ciphertexts (colored with low brightness) are irrelevant with the peak memory consumption.

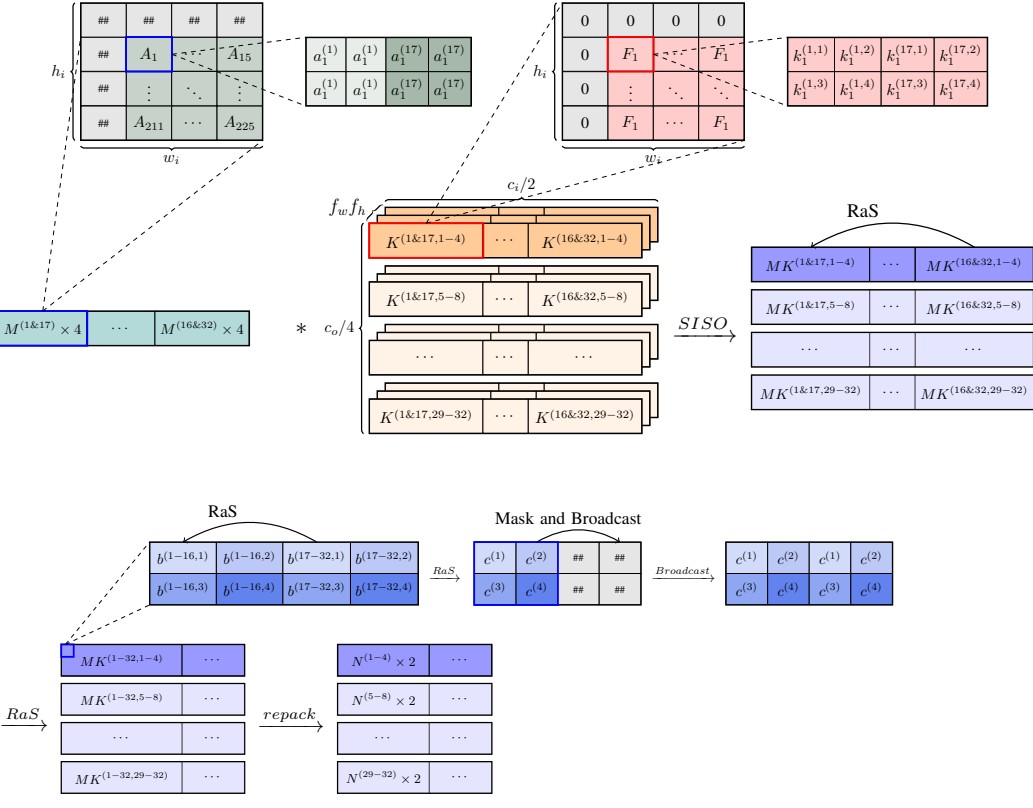

Figure 11: CAConv method with HP

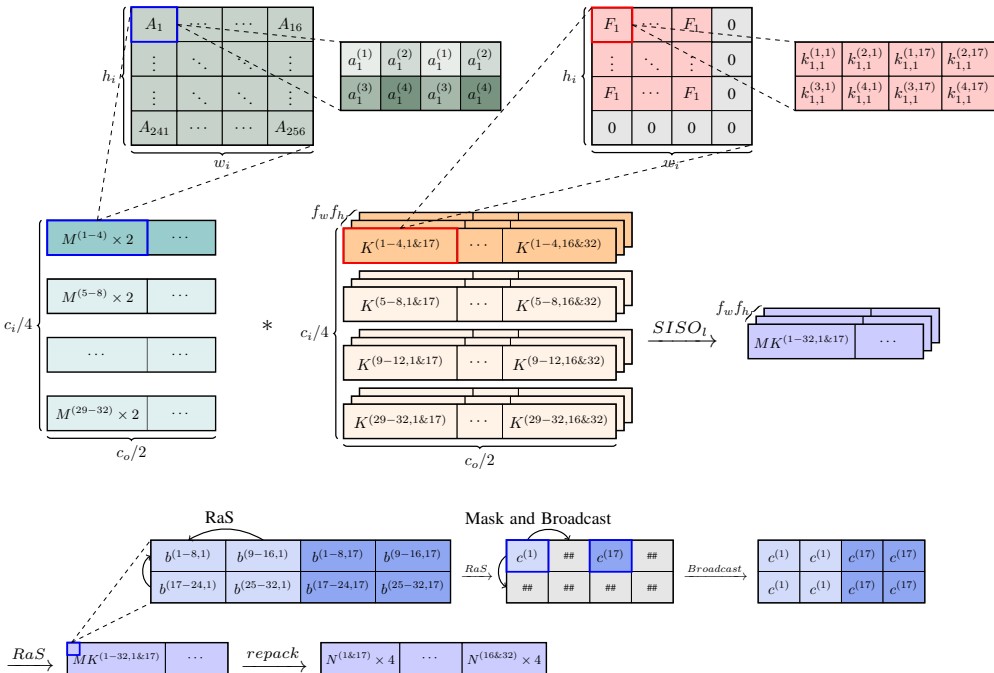

Figure 12: RAConv method with HP

