# OpenReview forum: "HyPHEN: A Hybrid Packing Method and Optimizations for Homomorphic Encryption-Based Neural Network "
_ICLR.cc/2023/Conference — Submitted to ICLR 2023_

### Official Review · Reviewer_WKRX · 2022-10-25

**Confidence:** 4
**Correctness:** 3
**Technical Novelty And Significance:** 3
**Empirical Novelty And Significance:** 3
**Recommendation:** 6

**Clarity, Quality, Novelty And Reproducibility:**

The RAConv and CAConv+RAConv ResNet basic block scheme seems novel and helpful for HE-DNN related research.
But the explanation needs to be made clear. Consider adding supplementary material to provide details. Please also provide sufficient explanations of the lazy-SISO and IR (Image Realign) and the experimental model structure of ResNet20/32/44/18.

**Strength And Weaknesses:**

Strengths

- The proposed packing method using CAConv and RAConv is novel in improving the performance of the heavy computation of CKKS FHE.

- RAConv has advantages over CAConv. The combination of CAConv+RAConv has advantages over only using CAConv.


Weaknesses

- Please report the information about the number of ciphertexts in your hybrid packing experiments. Also please report the amount of memory used for the comparison of pros and cons.

- This paper is hard to read in the explanation of the proposed RAConv and CAConv (Fig. 2) and the hybrid packing (Fig. 4). I have to read multiple passes of the section to understand.

- The details of "lazy-SISO" is not provided. It is mentioned multiple times in the paper but never explained how it works. What part of the computation is postponed and calculated when needed?


3.* Clarity, Quality, Novelty And Reproducibility Can you provide an evaluation of the quality, clarity and originality of the work?

**Summary Of The Paper:**

This paper proposes a hybrid packing method that can speed up the Fully Homomorphic Encryption (FHE) based neural network. Experiments are performed on speeding up the CKKS implementation of ResNet on CIFAR-10 image classification. The motivation is that, in the CKKS FHE computation of convolutions, the HE computation can be speed up by a dedicated design of repacking the values such that homomorphic rotation can be avoided, thus speed up the overall pipeline. Specifically, two new conv algorithms named RAConv and CAConv are proposed. The authors further extend the design of stacking RAConv and CAConv, in building a new scheme of ResNet basic block by  combining CAConv + RAConv. And with multiplex packing, the efficiency of the FHE CKKS pipeline is further improved. The experiments show that RAConv saves more time and the number of rotations over CAConv, and the combination of CAConv + RAConv is more efficient than only using CAConv.

**Summary Of The Review:**

The proposed RAConv and CAConv+RAConv Basic Block represent a contribution to the secure DNN inference research.

---

> ### Author Response · Authors · 2022-11-19
> **Response to Reviewer WKRX**
>
> We thank the reviewer for their time and efforts, as well as their valuable comments.
>
> **Q1**
> Please report the information about the number of ciphertexts in your hybrid packing experiments. Also please report the amount of memory used for the comparison of pros and cons.
>
> **A1**
> Thanks for the valuable comments to improve our paper, we added the number and total size of ciphertexts, and plaintexts in Appendix F. In Table 12 and 13, We reported the number of input ciphertexts and the peak number of ciphertexts during the computation. We also revised Figure 11 and 12 in our updated manuscript to reflect the actual computation process. As shown in Figure 11 and 12, the intermediate ciphertexts of the CAConv and input ciphertexts of the RAConv are larger than the input ciphertext of the CAConv. To avoid an increase in the number of ciphertexts, the operation on the input ciphertext continues until the number of ciphertexts shrinks again. While HyPHEN uses more ciphertexts due to duplication, the memory footprint compared to the prior work increases by up to 15-37% (7-15% if using the same FHE parameters). We see this memory overhead as affordable considering the latency benefits of our implementation.
>
> **Q2**
> The details of "lazy-SISO" is not provided. It is mentioned multiple times in the paper but never explained how it works. What part of the computation is postponed and calculated when needed.
> Consider adding supplementary material to provide details. Please also provide sufficient explanations of the lazy-SISO and IR (Image Realign) and the experimental model structure of ResNet20/32/44/18.
>
> **A2**
> We feel sorry for the reviewer for not providing the details of the lazy-SISO. We added the lazy-SISO section in Appendix E with Figure 9 following the suggestion of the reviewer. In lazy-SISO, rotations on the input ciphertexts are postponed. Thus after multiplying input ciphertexts with filter plaintexts, intermediate ciphertexts should be rotated to be accumulated. During this accumulation process, the intermediate ciphertexts with the same rotation index can be grouped to be accumulated first, and then rotated together, which reduces the amount of rotation by the group size. Lazy-SISO is beneficial when input channels, which are to be accumulated, are distributed in different ciphertexts as in the RAConv method.
>
> IR (Image Realign) only occurs using the implementation of [1]. As an input ciphertext stores multiple channels, rotation and sum (RaS) operation to accumulate channels results in the ciphertext with a single output channel. To recover the data representation which fills empty slots with channels, RaSed ciphertexts should be aggregated. During this process, RaSed ciphertexts should be masked and rotated before addition, which we dubbed the IR process. We updated the manuscript to have more explanations of IR following the reviewer’s suggestion.
>
> [1] Low-Complexity Deep Convolutional Neural Networks on Fully Homomorphic Encryption Using Multiplexed Parallel Convolutions, ICML 2022

---

### Official Review · Reviewer_JDDo · 2022-10-25

**Confidence:** 5
**Correctness:** 3
**Technical Novelty And Significance:** 3
**Empirical Novelty And Significance:** 3
**Recommendation:** 5

**Clarity, Quality, Novelty And Reproducibility:**

- The writing in the paper is relatively clear for the most part, expect for some experimental details that are unclear.
- The quality of the paper is high for the most part. The experiments are thorough, though there are a few relevant results that are missing.
- The proposed method and results cannot be reproduced without source code. And the paper does not comment on open sourcing the code for the research community. As such, there is limited scope for reproducing the results.

**Strength And Weaknesses:**

Strengths:

1. The paper identifies a bottleneck in implementing convolutional layers in CKKS, namely the large number of rotation operations. The paper proposes a new packing method that reduces the amount of expensive rotations.

2. The proposed convolution when coupled with low-degree polynomial approximation in AESPA leads to high-performance networks with better latency in ciphertext compared to multiplexed convolution. Although the source of the latency gains are probably elsewhere as described below.

Weaknesses:

1. The comparison to multiplexed convolution [2] (Lee et.al. 2022a) is not entirely fair. [2] seeks to approximate ReLU functions for directly employing them in pre-trained models, as opposed to training the models from scratch as is the case for HyPHEN. As such, the goals and operational settings of the two papers are not the same. HyPHEN needs models to be trained from scratch which is not always possible. This is an important distinction and that is ignored by the paper.

2. Encrypted inference in CNNs is bottlenecked in three respects,
 - (1) depth consumption of the circuit
 - (2) expense of bootstrapping operations
 - (3) cost of convolution operations.

Using low-degree polynomial approximations from AESPA mitigates (1). And, because of using low-degree approximations from AESPA, HyPHEN can use one less bootstrapping layer per residual block compared to [2], which helps mitigate (2). And since bootstrapping and polynomial approximations of ReLU consume the most levels and are the slowest parts, a majority of the latency gains in HyPHEN are probably from these two aspects as opposed to the proposed packing scheme.

Rebuttal Requests:
1. As far as I understand, the main difference from AESPA [1] is a convolution layer with fewer rotations. Yet, the performance of the ResNet models considered in this paper have improved from the corresponding ones in AESPA. Can you comment on what brought about this change, since changing the implementation of convolution should not affect the performance of the model itself?
2. In order to really understand the source of improved overall latency, can you provide a breakdown of the cost of each operation like Table 3 in [2]? Please report this for the proposed approach but with two versions of convolution, (a) multiplexed convolution, and (b) the proposed HP-CAConv+HP-RAConv? Or alternatively add another version of Table 4 where the convolution is multiplexed convolution as opposed to HP-CAConv+HP-RAConv. This should help delineate the source of latency improvements.
4. Reporting latency and claiming latency improvements is not adequate. Latency is affected by multiple system level factors, including number of cores, type and speed of each processor, throttling, other processing that maybe running etc. Measurements for the baselines and the proposed methods have to be performed under the exact same settings, which is hard to do, and must be repeated multiple times for reliability. Are the reported latency values for HyPHEN and the baseline measured under the same settings? Can you report mean and standard deviation over multiple runs?
5. Can you comment on the memory consumption for inference on encrypted data?

[1] AESPA: Accuracy Preserving Low-degree Polynomial Activation for Fast Private Inference, arXiv:2201.06699

[2] Low-Complexity Deep Convolutional Neural Networks on Fully Homomorphic Encryption Using Multiplexed Parallel Convolutions, ICML 2022

**Summary Of The Paper:**

The paper proposes a new packing scheme for improving the efficiency of convolution layers over homomorphically encrypted data. The paper first identifies that homomorphic rotations are the computational main bottleneck. Then a new data packing scheme is proposed to minimize the number of required rotations. The network architecture is based off of AESPA and evaluation is performed on CIFAR-10 using the HEAAN library. The proposed approach is able to reduce the computational complexity of ResNet models compared to multiplexed convolutions, the current state-of-the-art.

**Summary Of The Review:**

Overall, I am moderately positive about the paper; the new convolution layer certainly reduces the number of required rotations, which are computationally expensive. However, as pointed out above, there is missing information in the paper that does not allow the reader to understand the actual source of improvements, both from a performance and latency perspective.

**Update after Rebuttal:** I saw the rebuttal and the updates to the paper. The comparisons to the convolution in Lee 2022a are favorable from a latency perspective but at the price of higher memory requirements. So it is not clear if the proposed convolution is strictly better. The latency improvements over AESPA seem marginal. I am discarding the accuracy improvements over Lee 2022a since they are essentially a function of better plaintext training, which is equally applicable to Lee 2022a. Given the above, I am inclined to maintain my original rating of the paper.

---

> ### Author Response · Authors · 2022-11-19
> **Response to Reviewer JDDo**
>
> We thank the reviewer for their time and efforts, as well as their valuable comments.
>
> **Q1**:
> As far as I understand, the main difference from AESPA [1] is a convolution layer with fewer rotations. Yet, the performance of the ResNet models considered in this paper have improved from the corresponding ones in AESPA. Can you comment on what brought about this change, since changing the implementation of convolution should not affect the performance of the model itself?
>
> **A1**:
> Thank you for your valuable feedback. We have trained the ResNet model referring to the arXiv version of AESPA [1], which reports almost similar accuracies to our paper. While the training details of the network are the same through ResNet20 and 18 networks, we use a different but standard training method on ResNet 32 and 44. We added the training details to Appendix D in our updated manuscript as suggested by the reviewer.
>
> **Q2**:
> In order to really understand the source of improved overall latency, can you provide a breakdown of the cost of each operation like Table 3 in [2]? Please report this for the proposed approach but with two versions of convolution, (a) multiplexed convolution, and (b) the proposed HP-CAConv+HP-RAConv? Or alternatively add another version of Table 4 where the convolution is multiplexed convolution as opposed to HP-CAConv+HP-RAConv. This should help delineate the source of latency improvements.
>
> **A2**:
> We thank the reviewer for valuable suggestions to clarify our contributions. As the reviewer has suggested, we added the cost breakdown of each operation in Table 4. Furthermore, we added Table 6 to compare two versions of convolutions (namely, multiplexed convolution and the proposed convolution method). As the two implementations commonly use AESPA for activation with the same parameter set, the speedup is solely due to the proposed convolution method (HP-CAConv+HP-RAConv).
>
> **Q3**:
> Reporting latency and claiming latency improvements is not adequate. Latency is affected by multiple system level factors, including number of cores, type and speed of each processor, throttling, other processing that maybe running etc. Measurements for the baselines and the proposed methods have to be performed under the exact same settings, which is hard to do, and must be repeated multiple times for reliability. Are the reported latency values for HyPHEN and the baseline measured under the same settings? Can you report mean and standard deviation over multiple runs?
>
> **A3**:
> We are also grateful for the reviewer pointing out certain issues of reliability. All the experiments are conducted in the same environment without interference. We added the mean and the 95% confidence interval based on statistics running each experiment 20 times. The updated results in Table 3, 4, 6, 7, and 8 show that the runtimes are fairly stable.
>
> **Q4**:
> Can you comment on the memory consumption for inference on encrypted data?
>
> **A4**:
> We have added the details of memory consumption in Appendix F. We provide the size of a ciphertext, plaintext, evaluation key, and total keys (Table 11) and the required memory for each object (Table 12) while running each block of ResNet models. As Table 9 shows, plaintexts and evaluation keys take up most of the memory.
>
> More specifically, weight plaintexts in the convolution layers suffer from the highest memory expansion as the SISO-based convolution duplicates each element of the filter to the size of the images as shown in Figure 1 of our paper. While our implementation increases the number of ciphertexts compared to [2], the overall memory consumption increases by up to 15~37%, as shown in Table 12 and 13.
>
> Furthermore, we agree with the reviewer that ReLU approximation is beneficial as it does not require additional training. However, the size of the evaluation keys for running ResNet based on [2] is fairly high (146GB), which is not capable of GPU memory with the current HBM generation (80GB to the best of our knowledge). Alternatively, modifying the FHE parameter to have 8 levels to have lightweight evaluation keys would deteriorate the bootstrapping overhead even further, which is the reason we chose AESPA as our activation function. We see both activation functions have pros and cons; approximate ReLU entails a large overhead, and AESPA is lighter but restricted in some scenarios.
>
> [1] AESPA: Accuracy Preserving Low-degree Polynomial Activation for Fast Private Inference, arXiv:2201.06699
>
> [2] Low-Complexity Deep Convolutional Neural Networks on Fully Homomorphic Encryption Using Multiplexed Parallel Convolutions, ICML 2022

---

### Official Review · Reviewer_sH4P · 2022-10-29

**Confidence:** 5
**Correctness:** 3
**Technical Novelty And Significance:** 2
**Empirical Novelty And Significance:** 2
**Recommendation:** 5

**Clarity, Quality, Novelty And Reproducibility:**

The paper is clear and concise. The quality of the work is reasonable. Experiments are minimal but clear. Novelty is limited. Key training details and code are not available, which makes it difficult to assess reproducibility.


**Strength And Weaknesses:**

Strengths:
-	Figures explaining convolutional operations
-	Well-described problem statement

Limitations:

1) Main contributions of the paper, namely, RAConv and hybrid packing method have limited novelty. RAConv is not entirely different from CAConv and the hybrid packing method is the mixture of two existing packing methods. The resulting HyPHEN implementation can be described as a minor/incremental improvement over existing approaches.

2) Only the SISO case is considered. There is no mention about other scenarios (e.g., MIMO) and whether the proposed techniques can also be applied to other scenarios.

3) Results on larger datasets with higher resolution images (say 224 x 224 x 3) must be included. This is critical to determine if the proposed methods can make real-world private inference scenarios feasible.

**Summary Of The Paper:**

The paper presents a new HCNN implementation to perform private inference (PI) called HyPHEN. It proposes a replication-based convolution method (RAConv), which is innovatively alternated with the channel-aligned convolution method (CAConv). It also proposes a hybrid packing method, which is a combination of two existing packing methods: duplicate and multiplex packing. Both these methods reduce the overall number of homomorphic rotations and lead to 3-4 times lower latency compared to the baseline, with comparable accuracy based on ResNet architectures with RNS-CKKS implementation on the CIFAR-10 dataset.

**Summary Of The Review:**

The structure of the paper is good and easy to understand. Moreover, obtained results (e.g., speedup in throughput) are better than the state-of-the-art. However,  the novelty is limited and including experiments for complex settings (e.g., larger datasets) would enhance the paper.

---

> ### Author Response · Authors · 2022-11-19
> **Response to Reviewer sH4P**
>
> We thank the reviewer for their time and efforts, as well as their valuable comments.
>
> **Q1:**
> Main contributions of the paper, namely, RAConv and hybrid packing method have limited novelty. RAConv is not entirely different from CAConv and the hybrid packing method is the mixture of two existing packing methods. The resulting HyPHEN implementation can be described as a minor/incremental improvement over existing approaches.
>
> **A1:**
> We thank the reviewer for the valuable comments. In this paper, we showed that the replication of images in ciphertexts leads to favorable rotation reductions. To the best of our knowledge, duplicate packing (DP) has not been considered in the previous works. Specifically, [1] only considers packing a batch of data or multiple channels into a ciphertext. Batching multiple data leads to additional memory overhead, which is not feasible considering the GPU memory capacity. Packing slots using different channels leads to a significant rotation amount, which is proved through our paper. Thus we see our contribution as introducing a replication-based packing strategy (RAConv, DP and HP) to alternate between two data representations.
>
> **Q2:**
> Results on larger datasets with higher resolution images (say 224 x 224 x 3) must be included. This is critical to determine if the proposed methods can make real-world private inference scenarios feasible.
>
> **A2:**
> Thanks for your valuable suggestion on the necessity of results with higher-resolution images. We agree with the reviewer’s comment that the CIFAR-10 dataset is not sufficient to show the practicality of FHE-based private inference. Addressing this concern, we conducted additional experiments to evaluate the ImageNet dataset on the ResNet18 model. The total execution time takes 82 seconds in our GPU environment. We slightly modified the first pooling layer to average pool with kernel size 2x2 and stride 2. After the initial convolutional layer and average pooling layer, the number of intermediate ciphertexts is 8, and each ciphertext stores 8 channels with (m,d) = 1. We added Appendix G to explain the details of implementing the ImageNet version of ResNet18. We have not managed to update the accuracy results of the ImageNet version of ResNet18 within phase 1 of the review process. Furthermore, our CPU main memory is not sufficient to load weight plaintexts of the entire model and thus swap memory is used. We hope to report the refined latency and additional accuracy experiment during phase 2 of the review process.
>
> **Q3:**
> Only the SISO case is considered. There is no mention about other scenarios (e.g., MIMO) and whether the proposed techniques can also be applied to other scenarios.
>
> **A3:**
> Thanks for pointing out the other scenario. We have found the term MIMO used in the paper GALA [2]. Diagonal method used in GALA can also be applied to our method on ResNet18. However, using MIMO in ResNet20/32/44 incurs the slot utilization problem as the input feature maps are not large enough to fill a single ciphertext. As our paper shows that running a moderately large network such as ResNet18 using a high-resolution image on FHE is feasible, targeting other workloads with larger networks would be our future research direction.
>
> [1] Low-Complexity Deep Convolutional Neural Networks on Fully Homomorphic Encryption Using Multiplexed Parallel Convolutions, ICML 2022
>
> [2] GALA: Greedy ComputAtion for Linear Algebra in Privacy-Preserved Neural Networks, NDSS 2021

---

### Author Response · Authors · 2022-11-19
**General Response**

We appreciate all the reviewer’s constructive comments and positive feedback. We have addressed comments and suggestions through the individual responses and also revised the paper accordingly. To summarize the update of our manuscript, we added additional technical details of memory consumption, network architecture, parameters for each layer, and training details. We also elaborate on lazy-SISO to give more backgrounds about the algorithm. Most importantly, following the reviewer sH4P’s concern we show that our implementation can be applied to the ImageNet dataset.

We hope that we have sufficiently cleared all your concerns and we will be pleased to provide further information.

---

### Author Response · Authors · 2022-12-13
**Final Response**

We are sincerely grateful for the reviewer’s constructive feedback. While addressing the reviewer sH4P’s concern, we have improved ResNet18 inference on the ImageNet dataset to prove our implementation’s efficiency on the high-resolution images. However, swap memory is used during the program execution, and significantly deteriorates the runtime (Table 14). We have remedied the problem by fixing the memory allocator’s malfunction. Furthermore, we have improved memory transfer by using pinned (page-locked) memory, which has higher bandwidth than the default, pageable memory. In consequence, we get refined results for the ResNet18 inferences, where the total size of the weight plaintext exceeds the GPU memory capacity. In Appendix F, we described that our implementation of ResNet18 leverages multi-streaming concurrency, but we figured out that the compute and copy streams are executed sequentially at the kernel level. Our updated results with the new column ‘Memcpy’ are shown in the table below.

The runtime of the ResNet18 on the CIFAR-10 dataset is reduced to 10.16 (s) from 13.37 (s), solely due to the refined memory transfer. ResNet18 on the ImageNet dataset, in particular, takes 31.44 (s). Overall, we claim that our paper clearly shows the feasibility of the FHE inference with the realistic runtime on large networks and datasets for the first time. Thanks all.

|            | Im2Col | CAConv | RAConv | Bootstrap | Activation | Memcpy |  Etc |  Total |
|:----------:|:------:|:------:|:------:|:------:|:---------:|:----------:|:----:|:------:|
| runtime (s)  |  1.44  |  4.41  |  2.04  |    8.64   |    0.28 |  14.55   | 0.08 | 31.44 &pm; 1.5 |
| percent (%)   |   4.6  |  14.0  |   6.5  |    27.5   |     0.9  |  46.3   |  0.2 |   100  |

---

### Decision · Program_Chairs · 2023-01-20

**Decision:**

Reject

**Justification For Why Not Higher Score:**

The reviews are very confident and thorough, and I agree with the reviewers that the paper is just below the borderline.

**Justification For Why Not Lower Score:**

NA

**Metareview: Summary, Strengths And Weaknesses:**

The paper proposes a new HCNN called python. The methods of alternating convolutions (RA and CA) and the hybrid packing method reduces the overall number of rotations and reduces latency. It is felt that while the problem is interesting and important, the progress this represents over the prior work is relatively marginal.